Journal of Open Source Software (JOSS): design and first-year review

Smith Arfon M. arfon@stsci.edu 1
Niemeyer Kyle E. Kyle.Niemeyer@oregonstate.edu arfon.smith@gmail.com 2
Katz Daniel S. 3
Barba Lorena A. 4
Githinji George 5
Gymrek Melissa 6
Huff Kathryn D. 7
Madan Christopher R. 8
Cabunoc Mayes Abigail 9
Moerman Kevin M. 10 11
Prins Pjotr 12 13
Ram Karthik 14
Rokem Ariel 15
Teal Tracy K. 16
Valls Guimera Roman 17
Vanderplas Jacob T. 15
1 Data Science Mission Office, Space Telescope Science Institute , Baltimore , MD , United States of America
2 School of Mechanical, Industrial, and Manufacturing Engineering, Oregon State University , Corvallis , OR , United States of America
3 National Center for Supercomputing Applications & Department of Computer Science & Department of Electrical and Computer Engineering & School of Information Sciences, University of Illinois at Urbana-Champaign , Urbana , IL , United States of America
4 Department of Mechanical & Aerospace Engineering, The George Washington University , Washington , D.C. , United States of America
5 KEMRI—Wellcome Trust Research Programme , Kilifi , Kenya
6 Departments of Medicine & Computer Science and Engineering, University of California , San Diego, La Jolla , CA , United States of America
7 Department of Nuclear, Plasma, and Radiological Engineering, University of Illinois at Urbana-Champaign , Urbana , IL , United States of America
8 School of Psychology, University of Nottingham , Nottingham , United Kingdom
9 Mozilla Foundation , Toronto , Ontario , Canada
10 MIT Media Lab, Massachusetts Institute of Technology , Cambridge , MA , United States of America
11 Trinity Centre for Bioengineering, Trinity College, The University of Dublin , Dublin , Ireland
12 University of Tennessee Health Science Center , Memphis , TN , United States of America
13 University Medical Centre Utrecht , Utrecht , The Netherlands
14 Berkeley Institute for Data Science, University of California, Berkeley , CA , United States of America
15 eScience Institute, University of Washington , Seattle , WA , United States of America
16 Data Carpentry , Davis , CA , United States of America
17 University of Melbourne Centre for Cancer Research, University of Melbourne , Melbourne , Australia
Fox Edward
Electronic publication date: 2018 Feb 12
Publication date: 2018
Volume: 4
Electronic Location ID: e147
Received 2017 Oct 6; Accepted 2018 Jan 24
Copyright: ©2018 Smith et al.
Copyright year: 2018
Copyright holder: Smith et al.
License: This is an open access article distributed under the terms of the Creative Commons Attribution License, which permits unrestricted use, distribution, reproduction and adaptation in any medium and for any purpose provided that it is properly attributed. For attribution, the original author(s), title, publication source (PeerJ Computer Science) and either DOI or URL of the article must be cited.
License URL: https://creativecommons.org/licenses/by/4.0/

Keywords: Research software, Code review, Computational research, Software citation, Open-source software, Scholarly publishing

Funding: Alfred P. Sloan Foundation National Science Foundation ACI-1535065 1550224 National Institute of Health R01 GM123489 2017-2022 Leona M. and Harry B. Helmsley Charitable Trust 2016PG-BRI004 Gordon & Betty Moore Foundation Bill & Melinda Gates Foundation National Institute of Mental Health 1R25MH112480 This work was supported in part by the Alfred P. Sloan Foundation. Work by K E Niemeyer was supported in part by the National Science Foundation (No. ACI-1535065). Work by P Prins was supported by the National Institute of Health (R01 GM123489, 2017-2022). Work by K. Ram was supported in part by The Leona M and Harry B. Helmsley Charitable Trust (No. 2016PG-BRI004). Work by A Rokem was supported by the Gordon & Betty Moore Foundation and the Alfred P. Sloan Foundation, and by grants from the Bill & Melinda Gates Foundation, the National Science Foundation (No. 1550224), and the National Institute of Mental Health (No. 1R25MH112480). There was no additional external funding received for this study. The funders had no role in study design, data collection and analysis, decision to publish, or preparation of the manuscript.

==============================
This article describes the motivation, design, and progress of the Journal of Open Source Software (JOSS). JOSS is a free and open-access journal that publishes articles describing research software. It has the dual goals of improving the quality of the software submitted and providing a mechanism for research software developers to receive credit. While designed to work within the current merit system of science, JOSS addresses the dearth of rewards for key contributions to science made in the form of software. JOSS publishes articles that encapsulate scholarship contained in the software itself, and its rigorous peer review targets the software components: functionality, documentation, tests, continuous integration, and the license. A JOSS article contains an abstract describing the purpose and functionality of the software, references, and a link to the software archive. The article is the entry point of a JOSS submission, which encompasses the full set of software artifacts. Submission and review proceed in the open, on GitHub. Editors, reviewers, and authors work collaboratively and openly. Unlike other journals, JOSS does not reject articles requiring major revision; while not yet accepted, articles remain visible and under review until the authors make adequate changes (or withdraw, if unable to meet requirements). Once an article is accepted, JOSS gives it a digital object identifier (DOI), deposits its metadata in Crossref, and the article can begin collecting citations on indexers like Google Scholar and other services. Authors retain copyright of their JOSS article, releasing it under a Creative Commons Attribution 4.0 International License. In its first year, starting in May 2016, JOSS published 111 articles, with more than 40 additional articles under review. JOSS is a sponsored project of the nonprofit organization NumFOCUS and is an affiliate of the Open Source Initiative (OSI).

Introduction

Modern scientific research produces many outputs beyond traditional articles and books. Among these, research software is critically important for a broad spectrum of fields. Current practices for publishing and citation do not, however, acknowledge software as a first-class research output. This deficiency means that researchers who develop software face critical career barriers. The Journal of Open Source Software (JOSS) was founded in May 2016 to offer a solution within the existing publishing mechanisms of science. It is a developer-friendly, free and open-access, peer-reviewed journal for research software packages. By its first anniversary, JOSS had published more than a hundred articles. This article discusses the motivation for creating a new software journal, delineates the editorial and review process, and summarizes the journal’s first year of operation via submission statistics. We expect this article to be of interest to three core audiences: (1) researchers who develop software and could submit their work to JOSS, (2) those in the community with an interest in advancing scholarly communications who may appreciate the technical details of the JOSS journal framework, and (3) those interested in possibilities for citing software in their own research publications.

The sixteen authors of this article are the members of the JOSS Editorial Board at the end of its first year (May 2017). Arfon Smith is the founding editor-in-chief, and the founding editors are Lorena A. Barba, Kathryn Huff, Daniel Katz, Christopher Madan, Abigail Cabunoc Mayes, Kevin Moerman, Kyle Niemeyer, Karthik Ram, Tracy Teal, and Jake Vanderplas. Five new editors joined in the first year to handle areas not well covered by the original editors, and to help manage the large and growing number of submissions. They are George Githinji, Melissa Gymrek, Pjotr Prins, Ariel Rokem, and Roman Valls Guimera. (Since then, we added three more editors: Jason Clark, Lindsey Heagy, and Thomas Leeper.)

The JOSS editors are firm supporters of open-source software for research, with extensive knowledge of the practices and ethics of open source. This knowledge is reflected in the JOSS submission system, peer-review process, and infrastructure. The journal offers a familiar environment for developers and authors to interact with reviewers and editors, leading to a citable published work: a software article. The article describes the software at a high level, and the software itself includes both source code and associated artifacts such as tests, documentation, and examples. With a Crossref digital object identifier (DOI), the article is able to collect citations, empowering the developers/authors to gain career credit for their work. JOSS thus fills a pressing need for computational researchers to advance professionally, while promoting higher quality software for science. JOSS also supports the broader open-science movement by encouraging researchers to share their software openly and follow best practices in its development.

Background and Motivation

A 2014 study of UK Russell Group Universities (Hettrick et al., 2014) reports that ∼90% of academics surveyed said they use software in their research, while more than 70% said their research would be impractical without it. About half of these UK academics said they develop their own software while in the course of doing research. Similarly, a 2017 survey of members of the US National Postdoctoral Association found that 95% used research software, and 63% said their research would be impractical without it (Nangia & Katz, 2017).

Despite being a critical part of modern research, software lacks support across the scholarly ecosystem for its publication, acknowledgement, and citation (Niemeyer, Smith & Katz, 2016). Academic publishing has not changed substantially since its inception. Science, engineering, and many other academic fields still view research articles as the key indicator of research productivity, with research grants being another important indicator. Yet, the research article is inadequate to fully describe modern, data-intensive, computational research. JOSS focuses on research software and its place in the scholarly publishing ecosystem.

Why publish software?

Most academic fields still rely on a one-dimensional credit model where academic articles and their associated citations are the dominant factor in the success of a researcher’s career. Software creators, to increase the likelihood of receiving career credit for their work, often choose to publish “software articles” that act as placeholder publications pointing to their software. At the same time, recent years have seen a push for sharing open research software (Barnes, 2010; Vandewalle, 2012; Morin et al., 2012; Ince, Hatton & Graham-Cumming, 2012; Nature Methods Editorial Board, 2014; Prins et al., 2015).

Beyond career-credit arguments for software creators, publishing research software enriches the scholarly record. Buckheit and Donoho paraphrased Jon Claerbout, a pioneer of reproducible research, as saying: “An article about a computational result is advertising, not scholarship. The actual scholarship is the full software environment, code and data, that produced the result” (Buckheit & Donoho, 1995). The argument that articles about computational science are not satisfactory descriptions of the work, needing to be supplemented by code and data, is more than twenty years old! Yet, despite the significance of software in modern research, documenting its use and including it in the scholarly ecosystem presents numerous challenges.

Challenges of publishing software

The conventional publishing mechanism of science is the research article, and a researcher’s career progression hinges on collecting citations for published works. Unfortunately, software citation (Smith et al., 2016) is in its infancy (as is data citation (FORCE11, 2014; Starr et al., 2015)). Publishing the software itself and receiving citation credit for it may be a better long-term solution, but this is still impractical. Even when software (and data) are published so that they can be cited, we do not have a standard culture of peer review for them. This leads many developers today to publish software articles.

The developer’s next dilemma is where to publish, given the research content, novelty, length and other features of a software article. Since 2012, Neil Chue Hong has maintained a growing list of journals that accept software articles (Chue Hong, 2016). He includes both generalist journals, accepting software articles from a variety of fields, and domain-specific journals, accepting both research and software articles in a given field. For many journals, particularly the domain-specific ones, a software article must include novel results to justify publication.

From the developer’s point of view, writing a software article can involve a great deal of extra work. Good software includes documentation for both users and developers that is sufficient to make it understandable. A software article may contain much of the same content, merely in a different format, and developers may not find value in rewriting their documentation in a manner less useful to their users and collaborators. These issues may lead developers to shun the idea of software articles and prefer to publish the software itself. Yet, software citation is not common and the mostly one-dimensional credit model of academia (based on article citations) means that publishing software often does not “count” for career progression (Niemeyer, Smith & Katz, 2016; Smith et al., 2016).

The Journal of Open Source Software

To tackle the challenges mentioned above, the Journal of Open Source Software (JOSS) launched in May 2016 (Smith, 2016c) with the goal of drastically reducing the overhead of publishing software articles. JOSS offers developers a venue to publish their complete research software wrapped in relatively short high-level articles, thus enabling citation credit for their work. In this section we describe the goals and principles, infrastructure, and business model of JOSS, and compare it with other software journals.

Goals and principles

JOSS articles are deliberately short and only include an abstract describing the high-level functionality of the software, a list of the authors of the software (with their affiliations), a list of key references, and a link to the software archive and software repository. Articles are not allowed to include other content often found in software articles, such as descriptions of the API (application programming interface) and novel research results obtained using the software. The software API should already be described in the software documentation, and domain research results do not belong in JOSS—these should be published in a domain journal. Unlike most journals, which ease discoverability of new research and findings, JOSS serves primarily as a mechanism for software developers/authors to improve and publish their research software. Thus, software discovery is a secondary feature.

The JOSS design and implementation are based on the following principles:

• Other than their short length, JOSS articles are conventional articles in every other sense: the journal has an ISSN, articles receive Crossref DOIs with high-quality submission metadata, and articles are appropriately archived.

• Because software articles are “advertising” and simply pointers to the actual scholarship (the software), short abstract-length submissions are sufficient for these “advertisements.”

• Software is a core product of research and therefore the software itself should be archived appropriately when submitted to and reviewed in JOSS.

• Code review, documentation, and contributing guidelines are important for open-source software and should be part of any review. In JOSS, they are the focus of peer review. (While a range of other journals publish software, with various peer-review processes, the focus of the review is usually the submitted article and reviewers might not even look at the code.) The JOSS review process itself, described in Section ‘Peer review in JOSS’, was based on the on-boarding checklist for projects joining the rOpenSci collaboration (Boettiger et al., 2015).

Acceptable JOSS submissions also need to meet the following criteria:

• The software must be open source by the Open Source Initiative (OSI) definition (https://opensource.org).

• The software must have a research application.

• The submitter should be a major contributor to the software they are submitting.

• The software should be a significant new contribution to the available open-source software that either enables some new research challenge(s) to be addressed or makes addressing research challenges significantly better (e.g., faster, easier, simpler).

• The software should be feature-complete, i.e., it cannot be a partial solution.

How JOSS works

JOSS is designed as a small collection of open-source tools that leverage existing infrastructure such as GitHub, Zenodo, and Figshare. A goal when building the journal was to minimize the development of new tools where possible.

The JOSS web application and submission tool

The JOSS web application and submission tool is hosted at http://joss.theoj.org. It is a simple Ruby on Rails web application (Smith, 2016b) that lists accepted articles, provides the article submission form (see Fig. 1), and hosts journal documentation such as author submission guidelines. This application also automatically creates the review issue on GitHub once a submission has been pre-reviewed by an editor and accepted to start peer review in JOSS.

Figure 1 The JOSS submission page. A minimal amount of information is required for new submissions.

Open peer review on GitHub

JOSS conducts reviews on the joss-reviews GitHub repository (Smith, 2016a). Review of a submission begins with the opening of a new GitHub issue, where the editor-in-chief assigns an editor, the editor assigns a reviewer, and interactions between authors, reviewer(s), and editor proceed in the open. Figure 2 shows an example of a recent review for the (accepted) hdbscan package (McInnes, Healy & Astels, 2017). The actual review includes the code, software functionality/performance claims, test suite (if present), documentation, and any other material associated with the software.

Figure 2 The hdbscan GitHub review issue.

Whedon and the Whedon-API

Many of the tasks associated with JOSS reviews and editorial management are automated. A core RubyGem library named Whedon (Smith, 2016d) handles common tasks associated with managing the submitted manuscript, such as compiling the article (from its Markdown source) and creating Crossref metadata. An automated bot, Whedon-API (Smith, 2016e), handles other parts of the review process (such as assigning editors and reviewers based on editor input) and leverages the Whedon RubyGem library. For example, to assign the editor for a submission, one may type the following command in a comment box within the GitHub issue: @whedon assign @danielskatz as editor. Similarly, to assign a reviewer, one enters: @whedon assign @zhaozhang as reviewer (where the reviewer and editor GitHub handles identify them). The next section describes the review process in more detail.

Business model and content licensing

JOSS is designed to run at minimal cost with volunteer labor from editors and reviewers. The following fixed costs are currently incurred:

• Crossref membership: $275. This is a yearly fixed cost for the JOSS parent entity—Open Journals—so that article DOIs can be registered with Crossref.

• Crossref article DOIs: $1. This is a fixed cost per article.

• JOSS web application hosting (currently with Heroku): $19 per month

Assuming a publication rate of 100 articles per year results in a core operating cost of ∼$6 per article. With 200 articles per year—which seems possible for the second year—the cost drops to ∼$3.50 per article: (1) $275+$1×100+$19×12∕100=$6.03

(2) $275+$1×200+$19×12∕200=$3.51.

Submitting authors retain copyright of JOSS articles and accepted articles are published under a Creative Commons Attribution 4.0 International License (Creative Commons Corporation, 2016). Any code snippets included in JOSS articles are subject to the MIT license (Open Source Initiative, 2016) regardless of the license of the submitted software package under review, which itself must be licensed under an OSI-approved license (see https://opensource.org/licenses/alphabetical for a complete list).

Comparison with other software journals

A good number of journals now accept, review, and publish software articles (Chue Hong, 2016), which we group into two categories. The first category of journals include those similar to JOSS, which do not focus on a specific domain and only consider submissions of software/software articles: the Journal of Open Research Software (JORS, http://openresearchsoftware.metajnl.com), SoftwareX (https://www.journals.elsevier.com/softwarex/), and now JOSS. Both JORS (Chue Hong, 2017) and SoftwareX (Elsevier, 2017) now review both the article text and the software. In JOSS, the review process focuses mainly on the software and associated material (e.g., documentation) and less on the article text, which is intended to be a brief description of the software. The role and form of peer review also varies across journals. In SoftwareX and JORS, the goal of the review is both to decide if the article is acceptable for publication and to improve it iteratively through a non-public, editor-mediated interaction between the authors and the anonymous reviewers. In contrast, JOSS has the goal of accepting most articles after improving them as needed, with the reviewers and authors communicating directly and publicly through GitHub issues.

The second category includes domain-specific journals that either accept software articles as a special submission type or exclusively consider software articles targeted at the domain. For example, Collected Algorithms (CALGO, http://www.acm.org/calgo/) is a long-running venue for reviewing and sharing mathematical algorithms associated with articles published in Transactions on Mathematical Software and other ACM journals. However, CALGO authors must transfer copyright to ACM and software is not available under an open-source license—this contrasts with JOSS, where authors retain copyright and software must be shared under an open-source license. Computer Physics Communications (https://www.journals.elsevier.com/computer-physics-communications) and Geoscientific Model Development (https://www.geoscientific-model-development.net/) publish full-length articles describing application software in computational physics and geoscience, respectively, where review primarily focuses on the article. Chue Hong maintains a list of journals in both categories (Chue Hong, 2016).

Peer review in JOSS

In this section, we illustrate the JOSS submission and review process using a representative example, document the review criteria provided to authors and reviewers, and explain a fast-track option for already-reviewed rOpenSci contributions.

The JOSS process

Figure 3 shows a typical JOSS submission and review process, described here in more detail using the hdbscan package (McInnes, Healy & Astels, 2017) as an example:

1. Leland McInnes submitted the hdbscan software and article to JOSS on 26 February 2017 using the web application and submission tool. The article is a Markdown file named paper.md, visibly located in the software repository (here, and in many cases, placed together with auxiliary files in a paper directory).

2. Following a routine check by a JOSS administrator, a “pre-review” issue was created in the joss-reviews GitHub repository (hdbscan JOSS pre-review, 2016). In this pre-review issue, an editor (Daniel S. Katz) was assigned, who then identified and assigned a suitable reviewer (Zhao Zhang). Editors generally identify one or more reviewers from a pool of volunteers based on provided programming language and/or domain expertise1 .

The editor then asked the automated bot Whedon to create the main submission review issue via the command @whedon start review magic-word=bananas. (“magic-word=bananas” is a safeguard against accidentally creating a review issue prematurely.)

3. The reviewer then conducted the submission review (hdbscan JOSS review, 2016) (see Fig. 2) by working through a checklist of review items, as described in Section ‘JOSS review criteria’. The author, reviewer, and editor discussed any questions that arose during the review, and once the reviewer completed their checks, they notified the submitting author and editor. Compared with traditional journals, JOSS offers the unique feature of holding a discussion—in the open within a GitHub issue—between the reviewer(s), author(s), and editor. Like a true conversation, discussion can go back and forth in minutes or seconds, with all parties contributing at will. This contrasts with traditional journal reviews, where the process is merely an exchange between the reviewer(s) and author(s), via the editor, which can take months for each communication, and in practice is limited to one or two, perhaps three in some cases, exchanges due to that delay (Tennant et al., 2017).

Note that JOSS reviews are subject to a code of conduct (Smith & Niemeyer, 2016), adopted from the Contributor Covenant Code of Conduct (Ehmke, 2016). Both authors and reviewers must confirm that they have read and will adhere to this Code of Conduct, during submission and with their review, respectively.

4. After the review was complete, the editor asked the submitting author to make a permanent archive of the software (including any changes made during review) with a service such as Zenodo or Figshare, and to post a link to the archive in the review thread. This link, in the form of a DOI, was associated with the submission via the command @whedon set 10.5281/zenodo.401403 as archive.

5. The editor-in-chief used the Whedon RubyGem library on his local machine to produce the compiled PDF, update the JOSS website, deposit Crossref metadata, and issue a DOI for the submission (https://doi.org/10.21105/joss.00205).

6. Finally, the editor-in-chief updated the review issue with the JOSS article DOI and closed the review. The submission was then accepted into the journal.

Authors can also first submit a pre-submission inquiry via an issue in the main JOSS repository (Smith, 2016b) if they have questions regarding the suitability of their software for publication, or for any other questions.

Figure 3 The JOSS submission and review flow including the various status badges that can be embedded on third-party settings such as GitHub README documentation (Niemeyer, 2017b).

JOSS review criteria

As previously mentioned, the JOSS review is primarily concerned with the material in the software repository, focusing on the software and documentation. We do not ask authors to use their software in a research study or include research results in their article beyond as examples; submissions focused on results rather than software should be submitted to research journals. The specific items in the reviewer checklist are:

• Conflict of interest

– As the reviewer I confirm that I have read the JOSS conflict of interest policy and that there are no conflicts of interest for me to review this work.

• Code of Conduct

– I confirm that I read and will adhere to the JOSS code of conduct.

• General checks

– Repository: Is the source code for this software available at the repository URL?

– License: Does the repository contain a plain-text LICENSE file with the contents of an OSI-approved software license?

– Version: Does the release version given match the GitHub release?

– Authorship: Has the submitting author made major contributions to the software?

• Functionality

– Installation: Does installation proceed as outlined in the documentation?

– Functionality: Have the functional claims of the software been confirmed?

– Performance: Have any performance claims of the software been confirmed?

• Documentation

– A statement of need: Do the authors clearly state what problems the software is designed to solve and who the target audience is?

– Installation instructions: Is there a clearly-stated list of dependencies? Ideally these should be handled with an automated package management solution.

– Example usage: Do the authors include examples of how to use the software (ideally to solve real-world analysis problems)?

– Functionality documentation: Is the core functionality of the software documented to a satisfactory level (e.g., API method documentation)?

– Automated tests: Are there automated tests or manual steps described so that the function of the software can be verified?

– Community guidelines: Are there clear guidelines for third parties wishing to (1) contribute to the software, (2) report issues or problems with the software, and (3) seek support?

• Software paper

– Authors: Does the paper.md file include a list of authors with their affiliations?

– A statement of need: Do the authors clearly state what problems the software is designed to solve and who the target audience is?

– References: Do all archival references that should have a DOI list one (e.g., papers, datasets, software)?

Fast track for reviewed rOpenSci contributions

For submissions of software that has already been reviewed under rOpenSci’s rigorous onboarding guidelines (Ram, Ross & Chamberlain, 2016; Ram et al., 2017), JOSS does not perform further review. The editor-in-chief is alerted with a note “This submission has been accepted to rOpenSci. The review thread can be found at [LINK TO ONBOARDING ISSUE],” allowing such submissions to be fast-tracked to acceptance.

A Review of the First Year

By the end of May 2017, JOSS published 111 articles since its inception in May 2016, and had an additional 41 articles under consideration. Figure 4 shows the monthly and cumulative publication rates; on average, we published 8.5 articles per month, with some (nonstatistical) growth over time.

Figure 4 Statistics of articles published in JOSS since its inception in May 2016 through May 2017.

(A) Numbers of articles published per month, and (B) Cumulative sum of numbers of articles published per month. Data, plotting script, and figure files available (Niemeyer, 2017a).

Figure 5 shows the numbers of days taken for processing and review of the 111 published articles (i.e., time between submission and publication), including finding a topic editor and reviewer(s). Since the journal’s inception in May 2016, articles spent on average 45.5 days between submission and publication (median 32 days, interquartile range 52.3 days) The shortest review took a single day, for Application Skeleton (Zhang et al., 2016), while the longest review took 190 days, for walkr (Yao & Kane, 2017). In the former case, the rapid turnaround can be attributed to the relatively minor revisions needed (in addition to quick editor, reviewer, and author actions and responses). In contrast, the latter case took much longer due to delays in selecting an editor and finding an appropriate reviewer, and a multimonth delay between selecting a reviewer and receiving reviews. In other cases with long review periods, some delays in responding to requests for updates may be attributed to reviewers (or editors) missing GitHub notifications from the review issue comments. We have already taken steps to improve the ability of authors, reviewers, and editors to keep track of their submissions, including a prompt to new reviewers to unsubscribe from the main joss-reviews repository (Smith, 2016a) (to reduce unnecessary notifications) and a weekly digest email for JOSS editors to keep track of their submissions. In the future we may collect the email addresses of reviewers so we can extend this functionality to them.

Figure 6 shows the frequency of programming languages appearing in JOSS articles. Python appears the most with over half of published software articles (54), while R is used in nearly one-third of articles (29). We believe the popularity of Python and R in JOSS submissions is the result of (1) the adoption of these languages (and open-source practices) in scientific computing communities and (2) our relationship with the rOpenSci project.

Figure 5 Days between submission and publication dates of the 111 articles JOSS published between May 2016–May 2017. Data, plotting script, and figure file are available (Niemeyer, 2017a).

Figure 6 Frequency of programming languages from the software packages described by the 111 articles JOSS published in its first year.

Total sums to greater than 111, because some packages are multi-language. Data, plotting script, and figure file are available (Niemeyer, 2017a).

Each article considered by JOSS undergoes review by one or more reviewers. The set of 111 published articles have been reviewed by 93 unique reviewers. The majority of articles received a review by one reviewer (average of 1.11 ± 0.34), with a maximum of three reviewers. Based on available data in the review issues, on average, editors reached out to 1.85 ± 1.40 potential reviewers (at most 8 in one case) via mentions in the GitHub review issue. This does not include external communication, e.g., via email or Twitter. Overall, JOSS editors contacted 1.65 potential reviewers for each actual review (based on means).

Interestingly, the current reviewer list contains only 52 entries, as of this writing (reviewers.csv, 2017).2 Considering the unique reviewer count of 93, we clearly have reached beyond those that volunteered to review a priori. Benefits of using GitHub’s issue infrastructure and our open reviews include: (1) the ability to tag multiple people, via their GitHub handles, to invite them as potential reviewers; (2) the discoverability of the work so that people may volunteer to review without being formally contacted; (3) the ability to get additional, unprompted feedback and comments; and (4) the ability to find reviewers by openly advertising, e.g., on social media. Furthermore, GitHub is a well-known, commonly used platform where many (if not most) potential authors and reviewers already have accounts.

Figure 7 shows the numbers of articles managed by each of the JOSS editors. Editor-in-chief Arfon Smith stewarded the majority of articles published in the first year. This was somewhat unavoidable in the first three months after launch, as Smith served as the de facto sole editor for all submissions, with other members of the editorial board assisting. This strategy was not sustainable and, over time, we adopted the pre-review/review procedure to hand off articles to editors. Also, authors can now select during submission the appropriate editor based on article topic.

Figure 7 Numbers of articles handled by each of the JOSS editors.

Data, plotting script, and figure file are available (Niemeyer, 2017a).

Lastly, we analyzed the affiliations of the 286 authors associated with articles published in the first year. Figure 8 shows the number of authors per country; we represented authors with multiple affiliations in different countries using their first affiliation. Authors with no affiliation, or where we could not identify the country, are shown as “unknown.” From the articles published in the first year, approximately 48% of authors live in the United States and approximately 40% live in Europe (including Switzerland). The remaining 12% come from the rest of the world, most notably Australia (6.6%) and Canada (2.1%). Moving forward, we hope to receive submissions from authors in more countries that even better represent who develops research software around the world; one strategy to achieve this involves continuing to expand our editorial board.

Figure 8 Numbers of authors from a particular country.

Data, plotting script, and figure file are available (Niemeyer, 2017a).

In its first year, JOSS also developed formal relationships with two US-based nonprofit organizations. In March 2017, JOSS became a community affiliate of the Open Source Initiative (https://opensource.org), the steward of the open-source definition, which promotes open-source software and educates about appropriate software licenses. And, in April 2017, JOSS became a fiscally sponsored project of NumFOCUS (https://www.numfocus.org), a 501(c)(3) charity that supports and promotes “world-class, innovative, open source scientific computing.” Being associated with these two prominent community organizations increases the trust of the community in our efforts. Furthermore, as a NumFOCUS project, JOSS will be able to raise funding to sustain its activities and grow.

The Second Year for JOSS

Our focus for the second year will be on continuing to provide a high-quality experience for submitting authors and reviewers, and making the best use of the editorial board. In our first year, we progressed from a model where the editor-in-chief handled most central functions to one with more distributed roles for the editors, particularly that of ensuring that reviews are useful and timely. Editors can now select and self-assign to submissions they want to manage, while the editor-in-chief only assigns the remaining submissions. As JOSS grows, the process of distributing functions across the editorial board will continue to evolve—and more editors may be needed.

In the second year, we plan to complete a number of high-priority improvements to the JOSS toolchain. Specifically, we plan on automating the final steps for accepting an article. For example, generating Crossref metadata and compiling the article are both currently handled by the editor-in-chief on his local machine using the Whedon RubyGem library. In the future, we would like authors and reviewers to be able to ask the Whedon-API bot to compile the paper for them, and other editors should be able to ask the bot to complete the submission of Crossref metadata on their behalf. Other improvements are constantly under discussion on the JOSS GitHub repository (https://github.com/openjournals/joss/issues). In fact, anyone is able to report bugs and suggest enhancements to the experience. And, since the JOSS tools are open source, we welcome contributions in the form of bug-fixes or enhancements via the usual pull-request protocols.

Beyond roles and responsibilities for the editors, and improvements to the JOSS tools and infrastructure, we will take on the more tricky questions about publishing software, such as how to handle new software versions. Unlike traditional research articles that remain static once published, software usually changes over time, at least for maintenance and to avoid software rot/collapse (where software stops working because of changes in the environment, such as dependencies on libraries or operating system). Furthermore, because all potential uses of the software are not known at the start of a project, the need or opportunity arises to add features, improve performance, improve accuracy, etc. After making one or more changes, software developers frequently update the software with a new version number. Over time, the culmination of these changes may result in a major update to the software, and with many new contributors a new version might correspond to a new set of authors if the software is published. However, this process may not translate clearly to JOSS. The editorial board will accept a new JOSS article published with each major version or even a minor version if the changes seem significant enough to the editor and reviewer(s), but we do not yet know if this will satisfy the needs of both developers and users (corresponding to JOSS authors and readers, respectively).

The discussion about new software versions also generally applies to software forks, where software is copied and, after some divergent development, a new software package emerges. Similar to how we handle new software versions, the JOSS editorial board will consider publication of an article describing a forked version of software if it includes substantial changes from a previously published version. Authorship questions may be more challenging when dealing with forks compared with new versions, since forks can retain varying amounts of code from the original projects. However, while a version control history generally makes it easy to suggest people who should be authors, deciding on authorship can be difficult and subjective, and is therefore ultimately project-dependent. We prefer to leave authorship decisions to the projects, with discussion taking place as needed with reviewers and editors.

Conclusions

Software today encapsulates—and generates—important research knowledge, yet it has not entered the science publication ecosystem in a practical way. This situation is costly for science, through the lack of career progression for valuable personnel: research software developers. We founded JOSS in response to the acute need for an answer to this predicament. JOSS is a venue for authors who wish to receive constructive peer feedback, publish, and collect citations for their research software. By encouraging researchers to develop their software following best practices, and then share and publish it openly, JOSS supports the broader open-science movement. The number of submissions confirms the keen demand for this publishing mechanism: more than 100 accepted articles in the first year and more than 40 others under review. By the end of 2017, JOSS has published nearly 200 articles. Community members have also responded positively when asked to review submissions in an open and non-traditional format, contributing useful reviews of the submitted software.

However, we are still overcoming initial hurdles to achieve our goals. JOSS is currently not fully indexed by Google Scholar, despite the fact that JOSS articles include adequate metadata and that we made an explicit request for inclusion in March 2017 (see GitHub issue #130). Also, we may need to invest more effort into raising awareness of good practices for citing JOSS articles. That said, we have some preliminary citation statistics: according to Google Scholar, corner.py (Foreman-Mackey, 2016) and Armadillo (Sanderson & Curtin, 2016) have been cited the most at 116 and 79 times, respectively. Crossref’s Cited-by service—which relies on publishers depositing reference information—reports 45 and 28 citations for the same articles (JOSS Cited-by report, 2017). While most other articles have received no citations to-date, a few have been cited between one and five times. We have had at least two “repeat” submissions, i.e., submissions of a new version with major changes from a prior version. Clementi et al. (2017) published PyGBe-LSPR, a new version that added substantially new features over the original PyGBe of Cooper et al. (2016). Similarly, the software published by Sanderson & Curtin (2017) extended on (and cited) their earlier article (Sanderson & Curtin, 2016).

The journal cemented its position in the first year of operation, building trust within the community of open-source research-software developers and growing in name recognition. It also earned weighty affiliations with OSI and NumFOCUS, the latter bringing the opportunity to raise funding for sustained operations. Although publishing costs are low at $3–6 per article, JOSS does need funding, with the editor-in-chief having borne the expenses personally to pull off the journal launch. Incorporating a small article charge (waived upon request) may be a route to allow authors to contribute to JOSS in the future, but we have not yet decided on this change. Under the NumFOCUS nonprofit umbrella, JOSS is now eligible to seek grants for sustaining its future, engaging in new efforts like outreach, and improving its infrastructure and tooling.

Outreach to other communities still unaware of JOSS is certainly part of our growth strategy. Awareness of the journal so far has mostly spread through word-of-mouth and social networking (Tauber, 2016; Brown, 2017), plus a couple of news articles (Perkel, 2017; Moore, 2016). As of August 2017, JOSS is also listed in the Directory of Open Access Journals (DOAJ) (https://doaj.org/toc/2475-9066). We plan to present JOSS at relevant domain conferences, like we did at the 2017 SIAM Conference on Computational Science & Engineering (Smith et al., 2017) and the 16th Annual Scientific Computing with Python Conference (SciPy 2017). We are also interested in partnering with other domain journals that focus on (traditional) research articles. In such partnerships, traditional peer review of the research would be paired with peer review of the software, with JOSS taking responsibility for the latter.

Finally, the infrastructure and tooling of JOSS have unexpected added values: while developed to support and streamline the JOSS publication process, these open-source tools generalize to a lightweight journal-management system. The JOSS web application and submission tool, the Whedon RubyGem library, and the Whedon-API bot could be easily forked to create overlay journals for other content types (data sets, posters, figures, etc.). The original artifacts could be archived on other services such as Figshare, Zenodo, Dryad, arXiv, or engrXiv/AgriXiv/LawArXiv/PsyArXiv/SocArXiv/bioRxiv. This presents manifold opportunities to expand the ways we assign career credit to the digital artifacts of research. JOSS was born to answer the needs of research software developers to thrive in the current merit traditions of science, but we may have come upon a generalizable formula for digital science.

Additional Information and Declarations

Competing Interests

Author Contributions

Data Availability

1 Potential reviewers can volunteer via http://joss.theoj.org/reviewer-signup.html.

2 That reviewer list has since been replaced and is now available at https://docs.google.com/spreadsheets/d/1PAPRJ63yq9aPC1COLjaQp8mHmEq3rZUzwUYxTulyu78/edit?usp=sharing.

Daniel S. Katz is an Academic Editor for PeerJ CS. Abigail Cabunoc Mayes is an employee of Mozilla Foundation. Tracy K. Teal is an employee of Data Carpentry.

Arfon M. Smith conceived and designed the experiments, wrote the paper, prepared figures and/or tables, performed the computation work, reviewed drafts of the paper, edited submissions.

Kyle E. Niemeyer conceived and designed the experiments, analyzed the data, wrote the paper, prepared figures and/or tables, performed the computation work, reviewed drafts of the paper, edited submissions.

Daniel S. Katz and Lorena A. Barba, conceived and designed the experiments, wrote the paper, reviewed drafts of the paper, edited submissions.

George Githinji, Melissa Gymrek, Pjotr Prins, Ariel Rokem and Roman Valls Guimera reviewed drafts of the paper, edited submissions.

Kathryn D. Huff, Christopher R. Madan, Abigail Cabunoc Mayes, Kevin M. Moerman, Karthik Ram, Tracy K. Teal and Jacob T. Vanderplas conceived and designed the experiments, reviewed drafts of the paper, edited submissions.

The following information was supplied regarding data availability:

Niemeyer, Kyle (2017): JOSS first-year publication data and figures. figshare. https://doi.org/10.6084/m9.figshare.5147722.

Source code of JOSS, on GitHub: https://github.com/openjournals/joss.

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
