# Peer review of "Journal of Open Source Software (JOSS): design and first-year review"

_PeerJ Computer Science, doi:10.7717/peerj-cs.147_

## Round 0.1 · original submission · Minor Revisions

This article is viewed favorably by the reviewers, and should be published, after suitable revision.

Since it is an unusual submission, the reviewers have made many comments, though not always clearly indicating which are reflections and which call for changes. However, it really would help the article, and aid future readers, if as many as possible of those comments were to be addressed.

Reviewers 2 and 3 make specific comments about sections and paragraphs; those should certainly be attended to in a revision, as should the brief comments by Reviewer 1. Further, see some points below.

One repeated concern is about just having one reviewer for most of the works. This should be explained. How are conflicts of interest identified and addressed? How is having knowledgeable and skilled reviewers ensured? How is quality ensured if anyone can volunteer and review and that is the only checking done? How is correctness assessed in such cases? Has there been any review by others of the quality of the works accepted?

Another concern is support for readers. This seems a weakness and should be discussed, with plans articulated as to how it will be addressed. It would be nice to know who reads JOSS works, what they do after, if there are statistics on cites resulting, if there are repeat submissions, if newer submissions cite older ones, etc.

Another concern is about contextual issues like open science, being US centric, changes needed in promotion/advancement priorities in academia and industry, target audience, etc.

Another concern is about novelty of JOSS, literature review of related efforts, and understanding how JOSS works fit into the broader infrastructure. See for example below about Calgo. Further, how does it fit into the world of education (are submissions uses by teachers or students) or software engineering?

Another question is the reliance on Github and other parts of the assumed environment. The fact that most submissions are Python or R suggests that some discussion should be given as to broadening the universe from which submissions come. For example, with current interest in Jupyter notebooks growing, are submissions connected with such?

Please also see some marks in the manuscript copy attached.
Regarding the note “Calgo”, please see http://calgo.acm.org/ and http://toms.acm.org/algorithms-policy.cfm. Some discussion of this long-running algorithm venue seems necessary.

·

Basic reporting

no comment.

Experimental design

no comment.

Validity of the findings

no comment.

Additional comments

It might be nice to include some information on how reviewers for JOSS submissions are identified and/or selected. It seems quite an open review system and I wonder if the editors have protocols in place to deal with multiple unsolicited reviewers jumping onto a submission's review Github issue, potentially with conflicting opinions or biases in their experiences.

Given the JOSS review happens entirely via a Github issue and presumably most reviewers are often working with their own projects on Github, do the JOSS editors make use of any special customisations or additions to try and help JOSS review comments stand out from the general "noise" of Github notifications a reviewer is likely exposed to? Or any other thoughts from the editors to try and help make sure JOSS review notifications are not missed, as briefly mentioned as a reason for some delayed reviews.

·

Basic reporting

This manuscript describes the design of the Journal of Open Source Software and describes experiences after the first year since its inception. Let me start by mentioning that I fully support the ambition to publish open source software in articles that describe the software and which reviews some quality metrics and which makes it possible to cite software in scientific literature, with proper acknowledgements of the authors. I also appreciate the openness of the process and I congratulate the authors of this manuscript with the success of JOSS so far.

A paper like this can hardly be rejected on scientific merit, since it describes a process, and my comments are mostly an advice, sometimes a personal note and rarely a correction.

Section 1
Firstly, the intended audience is not entirely clear. There is quite a bit of technical jargon regarding software development in the paper. I think the jargon relates to the community of research software engineers and possibly bit wider than that (including this reviewer). The subject of publishing software in articles, making it citable and having the reward and recognition system in place for research software engineers in a wider academic context is indeed insufficiently appreciated. This paper is not going to respond to this discussion directly, so I suggest to make clear in the beginning that the targeted community is the already specialized research software community and readers who are likely to submit to this journal, rather than the wider academic community.

Related to the above, a bit wider discussion on the open science aspects would be appreciated (still within the scope of this particular article). I support the open software movement related to open science, but there are drawbacks and criticisms as well. Which aspects do you explicitly not resolve in this set up? The authors mention the ethics of open source in the beginning and describe at the end the issue of software development, but not the openness part. How does this journal contribute to the wider development of open science practices (open access clearly, but beyond open data to open software)? This is a subject which could be described in the introduction a bit more rather than taking it for granted

Comparable journals do exist, also journals that keep discussion papers open (the abstract suggests this is the first journal), e.g. those of the European Geophysical Union, including software and data journals such as GMD. It would be good to highlight more what the niche of JOSS is wrt these domain journals with similar goals.

It may also be relevant to refer and discuss instrumentation journals which have a similar goal. They don’t publish new scientific results, but rather describe instruments, calibrations etc.

A comment is that journal seems to be very US centered, with some exceptions, how does this reflect in article submissions? Is there a strategy in place to broaden it?

Section 1 ends without giving a summary what the journal entails. It is described further. It is about publishing software, but it doesn’t say how far it goes in that (does it include the code, documentation, tests or is it just a description?). A summary is needed for the reader.

Section 2.
While I agree with this section a number of comments:

The authors mention that actual scholarship is the full software environment, code that produced the data. This is a very limited view of science and scholarly research. It is a very technological view of scientific research only that suggests that the whole creative process of research is in software development. Also, it takes the pure academic view only, while 21st century research has tight connections to societal, economical and ecological challenges which is not reflected in a code. I highly disagree with the statement and I suggest to phrase it much more carefully. This is a personal note.

The motivation of this journal should not be the career of the research software engineers. It should be good scientific practice. As a side effect it could help the career development, but this problem relates more the reward and evaluation system in place in many academic institutions.
Having this journal may solve one aspect, but not the actual cause of lack of career perspectives of software engineers in the scientific community. This is to be addressed in the academic system, in particular the leadership of it.

That developers shun away from publishing a citable paper is not a strong argument. They should do so as part of good scientific practice since it contains an independent peer-review quality check, a time stamp and allows for referencing. The notion that a paper is also a time stamp recognizing the original contribution of the author to the scientific process could also be more clearly spelled out.

Note that there are quite a few domain specific journals that do not require new scientific results. From my own personal experience, as an editor of one of them, it is more a problem of culture. The reviewers focus a lot on the scientific results and find it hard to review papers purely based on methodologies and tools.

Section 3

'dressed up as ...' that sounds a bit weak.

It is unclear whether JOSS also checks whether the software is original, in the sense that it doesn’t re-invent a wheel. One of the goals of peer reviews of scientific papers is that the author is the first one to discover a result and puts a time stamp on it (see above). Does the journal allow any research software or does it help to avoid re-inventing the wheel? I miss this aspect about originality (see also section 4 on review criteria where it seems to be hidden in statement of need).

This section is a bit unclear on what is actually the target. On the one hand it is ‘ just’ advertising, on the other hand it mentions detailed code review.

If only high level functioning is reviewed, what is there actually to review in JOSS. How is quality and originality defined? So, how to determine impact and quality as peer review of scientific papers generally does? Does the code review have this role? Does this include coding standards, bench marking, checking for correctness of scientific result (in addition to code tests).

Section 3.4 compares only to generic research software journals. It would be good to compare to some domain journals as well.

Section 4/5

I find a minimum of one reviewer really low. What was the rationale to do so?

Given the review criteria, the scope of JOSS is much more than an advertisement. It is unclear though how much of the research application is required as well. It seems to focus primarily on the software aspects as such, not to the research application and the appropriateness of the software for that. This may be a deliberate choice, since more specialized domain journals pick this up, but it should be clearly stated.

I would be interested in two additional statistics: the country of affiliation of the first author (US centric?) and the distribution over the research application domains.

Section 6

The last paragraph is very long, a bit unclear and unsatisfactory in its ending. Is there a process foreseen to come to a choice to deal with this for JOSS?

Section 7

The first paragraph mentions the career perspective of RSEs, but this cannot be the reason to set up JOSS. It is set up because software is an essential part of the research process, and therefore it should be properly evaluated and published such that it is findable, with a measure of quality and originality.

I miss the open science perspective in this concluding section.

In conclusion, I have some remarks on the paper, but I strongly support JOSS as it is essential to publish research software and contribute to open science. I hope it will develop further in the near future!

Wilco Hazeleger

Experimental design

No specific comments

Validity of the findings

No specific comments

Additional comments

No specific comments

·

Basic reporting

The English is excellent, clear and unambiguous. The literature references provide the appropriate level of background and context.
The article is self-contained and uses a professional article structure. Data are available from a figshare article.

section 2 is a little repetitive (the final paragraph) as the case for garnering credit for software has been well made.

Experimental design

The purpose of the paper - which is to present the design of a new journal targetted at giving software "proxy" papers in a lightweught way and review its first year of operation - is clearly stated and presented.
The case for such a journal is convincingly made.
Section 3.1 (goals and principles) clearly presents the principles and justifications, and the mechanics of the Journal submission system and workflow are described in some detail in sections 3.2 and 4.

1. However, the design from the reader's experience is almost entirely missing. Journals have authors, editors/reviewers and readers, and it seems that the design for the reader experience is overlooked.
The Journal does not seem to have, for example, search functions and at over 246 articles (at time of access) a list without tags or labels is hard to navigate. Section 6 (the next year) focuses on automation of the submission process, including handling versions of previous submissions, but offers little in the way of features for readers.

It seems that the focus is on assigning a marker for software rather than a journal for seeking software articles. If this is an explicit design decision this should be stated.

The article focuses its design decisions on the editorial and reviewing ease of use. It would be instructive to have some feedback on the author and reader experience.

2. The review process primarily uses 1 reviewer - what is the confidence that this is sufficient? If the review process is primarily checklist based (as appears) with little judgement, then this could be enough.

Validity of the findings

The number of articles submitted and published is a clear indication of the justification and value of the Journal and its place in the ecosystem. The data presented is robust. Partnership with OSI and NumFOCUS is encouraging.

The review of the first year gives useful information. Some points that would improve the paper

1. the statistics given are from the point of view of the editorial team. What would be more interesting would be an analysis the submissions in the first year, by article topic, community served, geographic distributions (is this USA or European focused?) and so on. It is hard to tell from the user interface at http://joss.theoj.org/papers

2. As described in review section 2, there is little discussion of the reader experience. For example, I could not easily find if the Whedon RubyGem Library has it own article.

3. in addition to thoughts on versions, has the journal developed a strategy on software forking, in particular by different developer teams to the original developers?

4. although it is early days, is there any preliminary information on citation patterns for the articles? there is a hint only in "we may need to invest more effort into raising awareness of good practices for citing JOSS articles"

5. the reviewer counts and reviewer lists are different - are the reviewers rewarded? the implication is that the volunteers are not known.

All in all the article presents an innovative journal serving the software developer community, and I commend the the authors. The weaknesses are the sole focus on the editorial process and less on the reader and citation experience. I recognise that it is early days but some remarks on citation (the justifications given in section 2) would be welcome.

Additional comments

See sections above

---

## Round 0.2 · Minor Revisions

Many thanks for attending to all the comments provided through the review and editorial process of PeerJ CS. Note that the reviewers indicate you have addressed their concerns.

A small number of suggested changes should be considered to further improve the submission.

One point is to try to make the article easily understandable when read in the future, by avoiding expressions about time relative to the present. For example:
* p. 2, line 21, delete "currently";
* p. 2, lines 30-31, replace "JOSS recently . . . articles." with "More than one hundred articles were published in JOSS by its first anniversary.";
p. 16, lines 368, 369, 376 "next" -> "second"

Some other small edits include:
* p. 5, line 145: "simpler.)" -> "simpler)."
* p. 5, line 159: by -> with
* p. 9, line 237: contrasts -> contrasts with
* p. 11, line 270: url -> URL
* p. 11, line 291: software -> software,
* p. 18, line 459: figshare -> Figshare

Thanks again for working with PeerJ CS!

·

Basic reporting

The article is a special type of paper as it reviews the set up and experience of JOSS after 1 year. It is clearly written and the authors responded well to my comments. Otherwise, no comment.

Experimental design

See above, no comment.

Validity of the findings

See above, no comment.

Additional comments

Although I am still a bit concerned about the number of reviewers per article, I accept the arguments made and the role of the reviewers, which is different from those for a traditional scientific article. I congratulate the authors with the setup of JOSS and hope it will be a success.

·

Basic reporting

This is a resubmission - I will not repeat my previous comments, only commenting on the rebuttal and updates

The reporting is clearer. The scope of the journal, its primary aim and its relationship to other software journals is much clearer.

Experimental design

This is a resubmission - I will not repeat my previous comments, only commenting on the rebuttal and updates

The key point is the purpose of the journal - which is now much more clearly stated as a software developer publishing vehicle rather than a "traditional" journal where originality are part of the criteria and peer review is about quality gatekeeping rather than preparing to a reporting standard. This makes the various design and implementation decisions much more understandable.

The future features to embrace analysis, tagging, reader features, citation tracking and versioning are encouraging.

Validity of the findings

This is a resubmission - I will not repeat my previous comments, only commenting on the rebuttal and updates

Figure 8 and the new paragraphs on page 15 and page 17 have helped to address my comments on statistics and update strategies

Additional comments

This is a resubmission - I will not repeat my previous comments, only commenting on the rebuttal and updates

Clarifying the purpose and scope of the journal has considerably helped address some of the quirks of the design and reviewing methodology.

You are challenging not just the need for a forum for publishing new forms of research - like software - but also what it means to publish and to whose benefit it is. That will confuse traditionalists.

---

## Round 0.3 · accepted · Accept

Thanks for the speedy editing, addressing comments on the prior version. It will be great to see this published. Thanks for working with PeerJ CS!